# Stakeholders' Perceptions of New Digital Energy Management Platform in Municipality of Loulé, Southern Portugal: A SWOT-AHP Analysis

David Gago [1], Paula Mendes [1], Pedro Murta [1], Nuno Cabrita [1] and Margarida Ribau Teixeira [2],*

[1] Loulé City Hall, Praça da República, 8104-001 Loulé, Portugal; david.j.gh74@gmail.com (D.G.); paula.mendes@cm-loule.pt (P.M.); pedro.murta@cm-loule.pt (P.M.); nuno.cabrita@cm-loule.pt (N.C.)

[2] CENSE and Faculty of Sciences and Technology, University of Algarve, Campus de Gambelas, 8005-139 Faro, Portugal

\* Correspondence: mribau@ualg.pt; Tel.: +351-2-8980-0900

**Abstract:** This study aimed to develop a multi-stakeholder analysis to identify the best strategies for the integration of a new Digital Energy Management Platform (DEMP). The municipality of Loulé (South of Portugal) was used as a case study. A Strengths, Weaknesses, Opportunities, and Threats (SWOT) framework combined with an Analytical Hierarchy Process (AHP) framework and a TOWS Matrix was employed to analyse the stakeholder's perceptions to propose strategies for integrating the DEMP. Five focus stakeholder groups were involved. Results showed that stakeholders considered that the positive aspects of DEMP outweigh the negative aspects by approximately 36%. Strengths were ranked with 34.4%, Opportunities with 33.8%, Weaknesses with 20.2%, and Threats with 11.6%. The sequence of factors with the highest overall score by stakeholders was O1(12.7%) > S2(11.1%) > W2(7.4%) > T3(4.1%). Based on stakeholder perceptions, the most suitable strategies were those that use Strengths and Opportunities of the system (SO strategies), and strategies that take advantage of Opportunities while dealing with Weaknesses (WO strategies), achieving a prevalence compared with the other strategies of 34% and 27%, respectively. Therefore, the participation process involving stakeholders' groups in the implementation and monitoring of the DEMP provided an action plan and consensus capable of meeting the environmental and municipal energy management challenges.

**Keywords:** energy management; public services; sustainability; SWOT-AHP-TOWS

## 1. Introduction

As sustainability issues intensify worldwide [1], city managers, organizations and companies are being called to manage increasingly stressed resources with unprecedented efficiency [2–4]. Furthermore, sustainability has received greater relevance for society development. However, complex efforts are needed to maintain or create competitiveness and, at the same time, leadership improving the social and environmental impacts of human activities [5].

The 2030 Agenda for Sustainable Development (SDGs) is a global call to action for sustainable development, covering areas from poverty eradication and the provision of basic services to combat climate change and reduce inequalities [6]. The seventeen goals and their targets are defined as "integrated and indivisible" and to implement the 2030 Agenda their interconnected nature must be taken into consideration.

The energy represented by SDG7 aims to "guarantee access to affordable, reliable, sustainable and modern energy for all" [7]. The high demand for energy is irreversible, much derived from the improvement in the quality of life as well as the continuous increase in population [8]. Today municipalities face new challenges related to sustainable energy, introducing best environmental practices for energy efficiency in government buildings, facilities, transport, and waste and water management to make the energy system more

sustainable and reduce the negative impacts of climate change. The main goal to achieve a sustainable energy system in public administration is to develop strategies for energy efficiency [9].

Fekete et al. [10] emphasized that there should be tighter controls on energy efficiency and on the main sources of greenhouse gas emissions, such as freight transport, industry, buildings, and agriculture. However, an increasing number of researchers, policymakers, and practitioners are facing challenges to energy transition at an urban scale. In a recent study, two softwares were applied to quantify energy demand and sources to help prioritise actions for $CO_2$ reduction policies in urban areas [11].

In Portugal, there is some inefficient use of energy in the domestic, services and transport sectors, much of which is due to the increase in energy consumption over the past years [12]. In addition, large amounts of energy data are periodically generated in the public sector, and there is a need to store and process data on an intelligent energy platform for public management. The use of an intelligent Digital Energy Management Platform (DEMP) helps the public facility to monitor energy demand in real time, control billing, and analyse the suitability of options to rationalize consumption profiles, contract supply, and improve energy efficiency. The integration of remote-control functions in an intelligent platform and enables the punctual control of energy systems to eliminate superfluous consumption without compromising its functionality. The DEMP includes a graphs dashboard, making it possible to view information about costs, electricity consumption, and $CO_2$ emissions. These graphs can be changed according to the desired period and by the sector, subsector, or specific activity selected.

Eicker et al. [11] reported that urban platforms cover different domain areas, namely, in architecture, transportation conditions, and energy management, among others. These platforms mainly involve the collection and analysis of urban data from a wide variety of sources, such as indicators, municipal data records, information repositories or social media streams [13]. The concept of Big Data is emerging as new challenges appear in analysing, archiving, sharing, transferring, and processing large datasets across organizations [14]. Gandomi and Haider [15] emphasized that to handle large volumes of heterogeneous data stored within Big Data such as unstructured text, audio, and video formats, it is important to include broader and more efficient analytical methods. López-Robles et al. [16] stated that the use of intelligent systems is imperative in today's organizations, being important in the processes of collecting, analysing, interpreting and disseminating high-value data, as well as in the decision-making process. Marinakis and Doukas [8] stressed that the energy sector is already connected with internet technologies leading to efficient energy and environmental management.

Stakeholders are important in all complex system problems. Customers, users, clients, suppliers, employees, regulators, and team members of a system may be part of the solutions [17]. Environmental technologies pose uncertainties about their future development and implementation risks, and their acceptance is a gradual process by stakeholders [18]. In addition, Stojčetović et al. [19] stated that stakeholders can influence the decision, activity or result of a project. There are several stakeholder analysis tools that can be used for energy and environment management, including Strengths, Weaknesses, Opportunities, and Threats (SWOT) [20–23]. Haque et al. [22] used a well-defined SWOT analysis in combination with the Analytic Hierarchy Process (AHP) to identify barriers and Opportunities for electricity trade in Bangladesh by focusing on stakeholder perceptions. Results provided information on stakeholders' objectives to the implementation of energy measures and indicated that Weaknesses and Threats of the system were more pronounced than Strengths and Opportunities. One of the main strategies identified was the conclusion of a trilateral electricity trade agreement with joint development and implementation of cross-border electricity projects. Dhakal et al. [23] identified the main barriers and challenges for cross-border electricity trade and strategy development in Nepal using a Threats, Opportunities, Weaknesses and Strengths (SWOT-AHP-TOWS) approach. Based on this analysis, results indicated possible approaches that might be considered in Nepal to expand electricity

exports increasing demand for electricity in India and Bangladesh. De Las Casas et al. [24] strengthened the idea that management of energy resources must ensure sectoral policies as well as the roles performed by the different stakeholders and institutions involved (both public and private). Gottfried et al. [25] applied a SWOT-AHP-TOWS analysis to predict the behaviour of private investment in the Chinese biogas sector. They identified the Strengths, Weaknesses, Opportunities, and Threats (SWOT) of the stakeholders, then prioritised through the Analytical Hierarchy Process (AHP) and finally developed the TOWS Matrix. According to the results, the best strategies would be: financial promotion of customised biogas products; investment facility by insurance companies and pension funds; and new public-private partnerships and improve project cooperation by introducing professional guidance. Recently, Papapostolou et al. [26] carried out an AHP-SWOT approach together with the Fuzzy Technique of Preference Order by Similarity to the Ideal Solution (Fuzzy TOPSIS), with the aim of adopting the most suitable strategic plan to establish a successful energy cooperation that will create beneficial conditions for all parties involved. The main results pointed to the Opportunities for strategic energy cooperation between Europe and its neighbouring countries.

The aim of this work is to develop a multi-stakeholder SWOT-AHP-TOWS analysis to identify the most important criteria (internal and external factors) that can favour the integration of DEMP. Specifically, this study intends to (1) document stakeholders' perceived Strengths, Weaknesses, Opportunities and Threats for the integration and development of the DEMP; (2) carry out a comparative analysis on the perceptions of different stakeholder groups on energy management; and (3) involve stakeholders and provide strategies to integrate the new DEMP. This study is the first application of the SWOT-AHP-TOWS approach to the energy management in the public sector, to the best of the author's knowledge. The study takes place in the municipality of Loulé, southern Portugal.

## 2. Study Area

The municipality of Loulé is in the Algarve region, southern Portugal, with about 72,373 inhabitants [27]; it has an area of about 765 km$^2$ and a population density of 94.6 inhabitants/km$^2$. Algarve is a tourist region, so population has been increasing and is characterised by a very high seasonal variation driven mainly by tourism, the main economic activity. The seasonal population includes an additional 32,700 inhabitants during the period June to September [12,28].

Electricity consumption in the municipality of Loulé has undergone significant changes. The total amount of electricity consumed went from 604,483,994 kWh in 2001 to 615,950,279 kWh in 2015 [29]. Regarding the most recent period, industry is the main consumer, since it used 42.4% of the total of electricity consumed in the municipality [29]. Non-domestic and domestic consumption were responsible for 24% and 23.9%, respectively, of the total municipality electric energy consumption. There was a decrease in electricity consumption by the agricultural sector, which has become the sector with less relevance in energy consumption at the municipal level [29]. This growing trend in electricity consumption in the municipality of Loulé, together with the importance of having an updated smart grid, which allows the identification of the main sectors with high consumption, and the integration of new renewable energy sources, namely from solar energy already implemented, lead to the need to improve energy efficiency and create systems for the treatment of a large amount of data, such as an intelligent energy management platform.

The Sustainability and Natural Resources Division (DSRN) has the capacity to provide the municipality of Loulé with instruments that can meet the challenges currently faced in the area of energy efficiency. Some of the skills are: (i) develop the energy and environmental diagnosis; (ii) collaborate in the preparation of the Action Plan for Energy and Sustainability in the municipality of Loulé, and (iii) perform the tasks inherent to the energy manager [30].

Many decisions are made at the municipal level and a better connection between municipal decisionmakers and the general public is detectable during everyday life. Ad-

ditionally, there is a recognizable need to develop methods and tools that support the development of local energy systems [31].

## 3. Methodology

### 3.1. Overall Methodological Framework

According to the International Standardization Organization [32], energy consumption, energy use, and energy efficiency are related to the concept of energy performance. In order to effectively manage the energy performance of their facilities, systems, processes, and equipment, it is important for organisations to know how much energy is used and how much is consumed over time. Energy performance indicators are used to quantify results related to energy efficiency, use, and consumption [12].

The purpose of this study is to select which is the best strategy in the implementation of a DEMP, to make data management more efficient in the public service. As the objective of this study is to capture the stakeholders' perceptions in energy data management, a SWOT framework was combined with an AHP framework and a TOWS Matrix. SWOT provides the basic analysis to develop strategic planning and support decision-making, AHP assists in conducting SWOT more analytically and the TOWS Matrix allows the identification of decision strategies. The SWOT-AHP analysis is a method designed to assign weights to SWOT factors and categories, allowing decisionmakers to analyse a given situation more precisely and in more detail [33].

The effectiveness of the SWOT-AHP method depends strongly on the selection of stakeholders [23]. Therefore, a SWOT-AHP analysis was conducted using a questionnaire procedure with 25 representatives of five key stakeholder groups in the municipality of Loulé, as shown in Table 1. The chosen representatives have expertise in the energy data management sector and/or in decision-making positions and were available to answer the questionnaires.

**Table 1.** Stakeholder profiles.

| Number of Experts | Stakeholder Groups | Function Performed | Contribution Developed |
| --- | --- | --- | --- |
| 7 | Municipal Council of Loulé (MCL) | Public sector workers in the municipality of Loulé | Responsible for the implementation of the platform, allowing forecasts and implementation of energy efficiency measures |
| 6 | Academic Group (AG) | Researchers working in a higher education institution | Support for innovation and research |
| 6 | Computer Systems Company (ALGARDATA) | Specialized in the area of programming and database management | Importance in the automation of computer processes |
| 3 | Algarve Regional Energy and Environment Agency (AREAL) | Regional energy innovation, developing projects and energy certification | Collaborates in the execution of projects and implementation of measures to minimize the platform |
| 3 | Quinta do Lago Infrastructure Company, EM (INFRAQUINTA) | Responsible for infrastructure management in the public sector | Gathers a set of skills and knowledge in the field of energy efficiency, playing an important role in the application of the platform |

The study methodology is shown in Figure 1. In Stage 1, the study analysed prior research (i.e., literature review) on the energy management and collected data (detailed in Section 3.2). Then, Stage 2 is where the SWOT Analysis was defined. A questionnaire was provided to stakeholders to select SWOT factors. Based on the stakeholders' perception, the internal (Strength and Weakness) and external (Opportunities and Threat) factors with

relative importance to DEMP were selected to formulate the SWOT-AHP analysis model (Section 3.3). Based on the selected factors, an AHP survey was conducted to measure the global priority of each factor, in Stage 3 (Section 3.4). Once the SWOT-AHP results were calculated, a TOWS Matrix was developed to analyse and prioritize the policies, and to develop alternative strategies, which consisted of Stage 4 (Section 3.5).

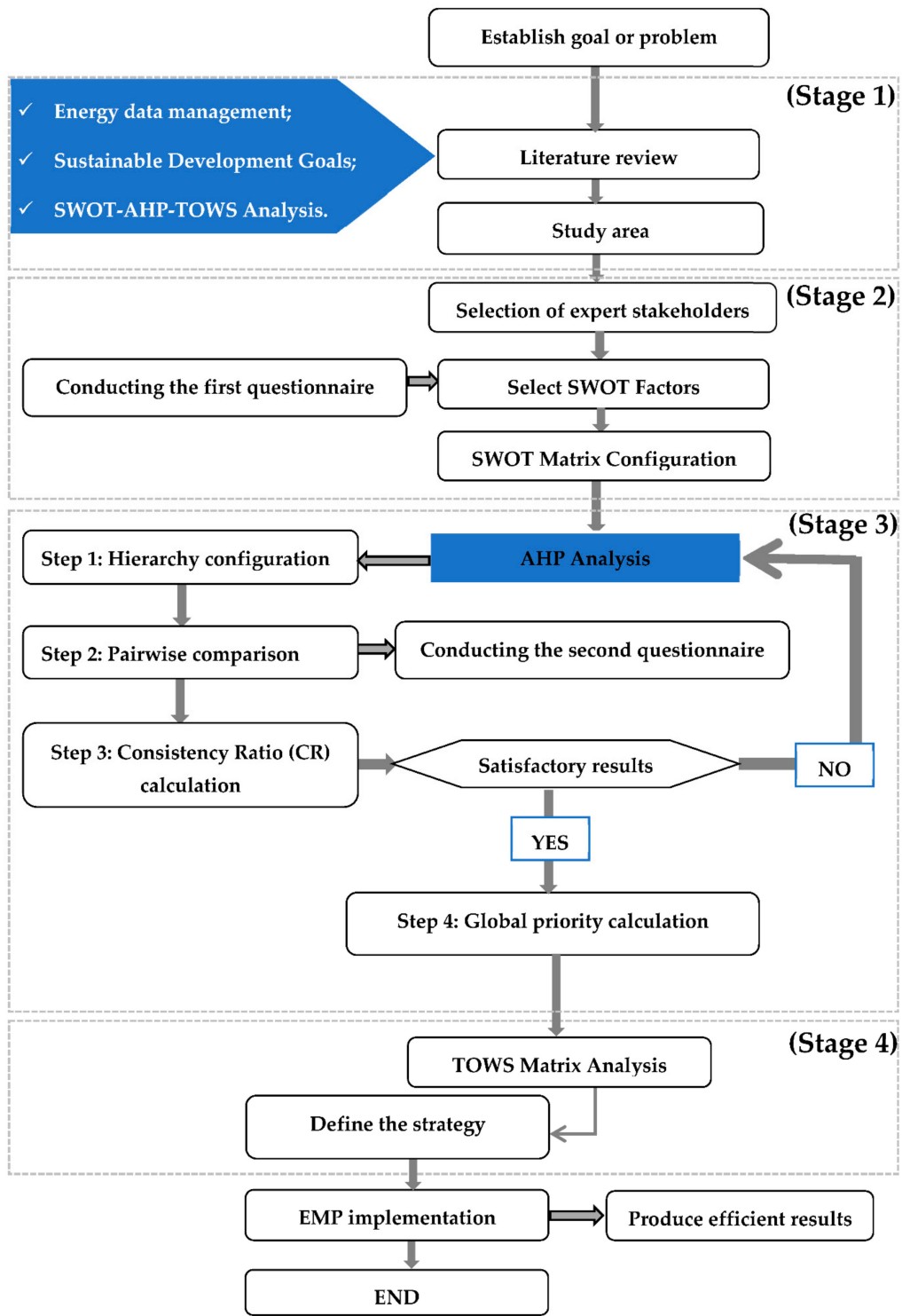

**Figure 1.** Methodology structure used in the study. Stage 1, preparation and data collection; Stage 2, SWOT analysis; Stage 3, AHP analysis; Stage 4, TOWS Matrix analysis.

### 3.2. Preparation and Data Collection—Stage 1

One of the main problems in the municipalities is the paucity of an organized and historical energy database. These data are needed to take appropriate measures to reduce costs, electricity consumption, and $CO_2$ emissions. Policymakers need to move quickly to keep pace with digital technology changes, providing Opportunities for energy transitions, allowing the development of flexible power systems [34]. The Sustainable Development Goals (SDGs) agenda emphasizes the need to develop targets and objective indicators to monitor progress, implement strategies, allocate resources, and increase the accountability of stakeholders [1].

Regarding data collection, first the responsibilities of each team member were assigned, then a pre-selection of relevant context information was made, and finally the data and the relevant factors for the study were collected. This research followed the SWOT-AHP analysis as outlined in by Kurttila et al. [33], Gottfried et al. [25] and Uddin et al. [35].

### 3.3. SWOT Analysis—Stage 2

The SWOT analysis identifies the research problem and determines the strategic goals and objectives of a study [36]. Here, the internal factors consist of the Strengths (positive) and Weaknesses (negative) of an energy management platform, while the external factors are the Opportunities (positive) and Threats (negative) that exist to implement DEMP. This analysis allows to maximize the Strengths (S) and Opportunities (O), but it can simultaneously minimise the Weaknesses (W) and Threats (T) [36].

The final SOWT matrix was developed based on the literature review on the topic and a list of SWOT factors developed by stakeholders during the first questionnaire. The first questionnaire consisted of four parts, and short answers. This questionnaire was conducted during May 2021. The number of participants (25) were considered sufficient to obtain expert opinions on the topic [23]. Subjective questions were used to identify important factors that could influence the integration of the DEMP [37].

### 3.4. AHP Analysis—Stage 3

The importance of each SWOT factor cannot be determined using a conventional SWOT analysis [19]. This can be overcome by applying a quantitative approach to the information generated from the SWOT analysis, such as integrating the Analytic Hierarchy Process (AHP). AHP is one of the most widely applied Multi-Criteria Decision Analysis (MCDA) tools as it incorporates some of the following topics: (1) ability in analysing conflicting preferences [38]; (2) allows quantifying and comparing of qualitative opinions of stakeholders [38]; (3) provides the ability to transform intangible factors into numerical values [39]; and (4) facilitates systematically evaluating the weights of selected factors in pairs through a range [40].

In Serbia, Stojčetović et al. [19] used the SWOT-AHP method to determine the actual energy (electricity) situation in the municipality of Štrpce and define and evaluate possible strategies to improve energy security, and Solangi et al. [41] developed several strategies for sustainable energy planning in Pakistan.

The AHP method is composed of four steps [42,43]. In step 1, a hierarchy is built based on the selected SWOT factors and categories (Section 3.4.1). In step 2, experts perform pairwise comparisons to derive the priorities of categories and factors (Section 3.4.2). In step 3, consistency ratios (CRs) are considered to identify consistency of second questionnaires that are appropriate for analysis (Section 3.4.3). In step 4, the global priority of SWOT factors and categories are calculated (Section 3.4.4). Finally, the results of the study are presented in Section 4.

#### 3.4.1. Hierarchy Configuration

The SWOT-AHP hierarchy is conducted based on the factors selected within the SWOT categories. The SWOT-AHP method has three levels, as shown in Figure 2. The top level involves the suggestion of strategies for the development of DEMP. The second level is

based on the four SWOT categories (criteria) and the third level consists of four factors (sub-criteria) for each category in the second level.

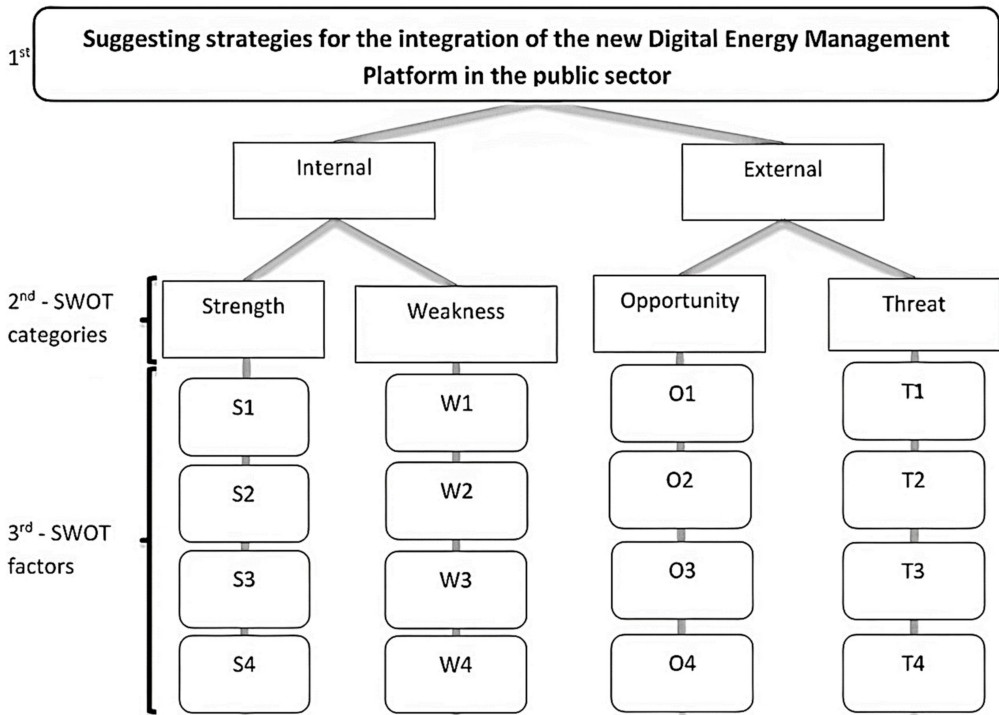

**Figure 2.** Hierarchical structure of the SWOT-AHP factors.

3.4.2. Pairwise Comparisons

In order to qualitatively assess the associated SWOT factors, stakeholders were asked to complete a second questionnaire during June 2021, and data were collected from the 25 experts. In each SWOT category and for each stakeholder group, the factor with the highest priority was identified and comparison between the factors with the highest priority score was conducted. This was developed to estimate the overall priority of the different factors in relation to each other. Throughout the analysis, consistency scores ≤0.1 were maintained. Microsoft Excel 2016 was used to simulate the priorities based on the pairwise comparisons for each stakeholder group and to verify their consistency.

A response from each expert was based on this partiality by balancing two given factors. Each category and factor were weighted through pairwise comparisons series using a 1–9 scale [40], presented in Table 2.

**Table 2.** Relative importance scale (adapted from [41–44]).

| Point Scale | Reciprocal | Definition of Preference Judgements |
|:---:|:---:|---|
| 1 | 1 | The contribution of both factors is equal |
| 3 | 1/3 | One factor is slightly favoured over another |
| 5 | 1/5 | One factor is strongly favoured over another |
| 7 | 1/7 | The factor is strongly favoured and its prevalence is demonstrated in practice |
| 9 | 1/9 | Importance of one over another affirmed on the highest possible order |
| 2,4,6,8 | 1/2,1/4,1/6,1/8 | Used to represent compromise between the priorities listed above |

Six pairwise comparisons were made for the SWOT categories, followed by 24 pairwise comparisons among the four factors from each of the four categories, as shown in Table 3. The pairwise comparison questions are constructed as shown in Figure 2.

**Table 3.** Two stages of pairwise comparison sets of SWOT factors.

| Stage | | | Pairwise Comparisons | |
|---|---|---|---|---|
| 1 | 6 | Categories | (S:W), (S:O), (S:T) (W:O), (W:T) (O:T) | |
| 2 | 24 | Factors | (S1:S2), (S1:S3), (S1:S4) (S2:S3), (S2:S4) (S3:S4) (W1:W2), (W1:W3), (W1:W4) (W2:W3), (W2:W4) (W3:W4) | (O1:O2), (O1:O3), (O1:O4) (O2:O3), (O2:O4) (O3:O4) (T1:T2), (T1:T3), (T1:T4) (T2:T3), (T2:T4) (T3:T4) |

According to Kurttila et al. [33], when making comparisons many questions are asked, such as (1) which of the two factors compared has a greater Strength (for Figure 3 Strength vs. Opportunity, Weakness, or Threat); and (2) how much greater is the Strength is.

**Figure 3.** An example of a pairwise comparison question (example for the Strength factor).

After collecting the responses provided by the experts, the arithmetic mean was used to equilibrate the responses received. The weight, which is the value of relative importance in the decisionmaker's opinion, is then produced and further calculated using the eigenvalue method.

### 3.4.3. CR Calculation

For a more precise analysis and to determine the consistency of the data, the CR value was calculated. This calculation used the pairwise comparison from the questionnaire analysis. Questionnaires with CR values equal or higher than 0.1 were not included in posterior analysis [45].

The consistency checking procedure was as follows. First, to verify consistency, each pairwise comparison result must be transformed into a matrix, as shown in Equation (1).

This matrix is formed based on the assumption that if *A* is preferred X times more than B, B should be preferred 1/X times more than *A* [46].

$$A = (a_{ij}) = \begin{bmatrix} 1 & \frac{W1}{W2} & \cdots & \frac{W1}{Wn} \\ \frac{W2}{W1} & 1 & \cdots & \frac{W2}{Wn} \\ \vdots & \vdots & \cdots & \vdots \\ \frac{Wn}{W1} & \frac{Wn}{W2} & \vdots & 1 \end{bmatrix} \tag{1}$$

In matrix *A*, rows represent the relative weight of each factor to the others, when *i* = *j*, $a_{ij}$ = 1. When the transpose of the vector of weights *w* is multiplied by matrix *A*, a vector represented by $\lambda_{max}w$ is obtained, where $\lambda_{max}$ is the largest eigenvalue of matrix *A* and *w* is the transpose of the vector of weights. Equation (2) can be written as:

$$(A - \lambda_{max}I)W = 0 \tag{2}$$

where *I* is the identity matrix. The largest eigenvalue, $\lambda_{max}$, is equal to or greater than *n* or the number of rows or columns in the matrix *A* [44]. If there is greater consistency among responses, $\lambda_{max}$ is closer to *n*. If all responses are perfectly consistent then $\lambda_{max}$ equals *n* [33,44]. A CI for each matrix is then estimated using the Equation (3):

$$CI = \frac{\lambda_{max} - n}{n - 1} \tag{3}$$

The *CR* is calculated using Equation (4) to determine the degree of consistency and depends on the calculation of the consistency index (*CI*) (see Equation (3)) and the random index (*RI*) [33]. The random index (*RI*) value refers to the random index proposed by Saaty [44]:

$$CR = \frac{CI}{RI} \tag{4}$$

The *RI* is the mean value of the *CI* of random generated comparison matrices employing Saaty's preference scale. Table 4 shows the value of the *RI* for matrices of order (*n*) from 1 to 10.

**Table 4.** Random consistency index (RI) [42].

| *n* | 1 | 2 | 3 | 4 | 5 | 6 | 7 | 8 | 9 | 10 |
|-----|------|------|------|------|------|------|------|------|------|------|
| RI | 0.00 | 0.00 | 0.58 | 0.90 | 1.12 | 1.24 | 1.32 | 1.41 | 1.45 | 1.49 |

Saaty [40] suggested that the value of CR should be less than or equal to 0.10. Inconsistency of 10 percent implies that there is a 10% chance that the decisionmaker will answer the questions in a random manner. If the CR is much more than 0.1, an inconsistency of pairwise comparisons results is considered [40].

3.4.4. Global Priority

After pairwise comparisons were made between each factor within the SWOT categories, pairwise comparisons between each category were made using a similar approach. According to the number of valid questionnaires that were consistent, the final weights were calculated using the help of the weighted mean of each factor and SWOT category. In a typical SWOT-AHP analysis, an overall priority for each factor would be calculated using Equation (5) [47].

$$\begin{aligned} Global\ priority\ of\ factor\ ij \\ = Priority\ value\ of\ factor\ ij \times Scaling\ value\ of\ SWOT\ category \end{aligned} \tag{5}$$

where *i* is equal to the number of factors in a SWOT category and *j* is equal to 4 (Strengths, Weaknesses, Opportunities, and Threats). The scale value for each SWOT category was calculated using the eigenvalue method that was described to determine the priority values of the factors. This formula emphasises that all SWOT categories are not dependent on each other.

### 3.5. TOWS Matrix Analysis

The TOWS analysis has been applied widely to develop strategies based on a previous SWOT-AHP analysis [22,48]. This analysis can also be used in developing the tactics needed to implement strategies and to look for more specific actions to support these tactics [49]. For a TOWS analysis, Threats and Opportunities are examined first, then Weaknesses and Strengths. After creating a list of Threats, Opportunities, Weaknesses, and Strengths, strategies are formulated to find out how advantages of Opportunities and minimization of Threats can be developed by exploiting Strengths and overcoming Weaknesses [23].

According to Weihrich [49] and Papapostolou et al. [26], TOWS involves four strategies:

- The Weakness-Threat (WT) strategy that attempts to reduce the impact of Threats by considering Weaknesses (min-min);
- The Weakness-Opportunity (WO) strategy takes into account Weaknesses to obtain the benefits of Opportunities (min-max);
- The Strengths-Threat (ST) strategy is based on the principle of making good use of strengths to eliminate or reduce the impact of Threats (max- min);
- The Strengths-Opportunity (SO) strategy is based on the principle of making good use of Opportunities through existing Strengths (max-max).

Currently, a combination of SWOT, AHP, and TOWS is popular in many sectors including economic [50], biofuels [48], and electricity [23]. However, the application of SWOT–AHP–TOWS analysis in the energy management sector is still limited.

The results were organized into a table, and graphs are also created based on the weights. Strategies were then constructed for the factors with the eight weights equal to or greater than 0.060, to limit the study and allow the development of strong strategies for the implementation of DEMP.

## 4. Results and Discussion

### 4.1. SWOT Analysis and CR Determination

The total twenty-five questionnaires were considered, all of which with eligible responses. The SWOT factors (sub-criteria) were defined in significant categories based on the literature review and analysis of the factors listed by the experts. Following the set of the preliminary list of decision factors, these factors were categorized into sixteen major factors, four of which were placed in each SWOT category (Table 5). The number of pairwise comparisons in AHP grows exponentially as the factors increase, although it is beneficial to consider as many factors as possible. A maximum of four key factors was considered in each SWOT category to keep pairwise comparisons at a manageable level.

**Table 5.** Selected SWOT factors.

| SWOT Categories | | | Factors |
|---|---|---|---|
| S | STRENGTHS | S1 | Early detection of anomalies, automatic suggestions of improvements and optimization of consumption |
| | | S2 | Optimization of consumption and quantification of $CO_2$ emissions |
| | | S3 | Production of reasoned and essential information for decision-making |
| | | S4 | Potential better use of energy resources |
| W | WEAKNESSES | W1 | Difficulty in the interconnection of data from the energy supply company with the georeferencing of electricity meters |
| | | W2 | Lack of specialized human resources |
| | | W3 | Budget constraints |
| | | W4 | Difficulty in interconnecting with other energy management systems |
| O | OPPORTUNITIES | O1 | Existence of technologies capable of monitoring energy consumption in real time |
| | | O2 | Increasing trend in electricity consumption |
| | | O3 | Existence of a legal framework that places particular emphasis on the implementation of energy efficiency measures and renewable energy sources |
| | | O4 | Characteristics of the Loulé territory that favour the rational use of energy and the use of renewable energy sources |
| T | THREATS | T1 | Context of economic crisis |
| | | T2 | Low level of adherence to awareness campaigns about new energy management technologies among the population |
| | | T3 | Difficulties in accessing investment and new energy management technologies |
| | | T4 | High cost associated with the implementation of electrical equipment and use of renewable energy sources |

Source: authors' own projection based on the results of the literature review and the experts' answers to the questionnaire.

Table 5 denotes the Strength category as S1 to S4, the Weakness category as W1 to W4, the Opportunity category as O1 to O4, and the Threat category as T1 to T4.

Only questionnaires with CR values less than 0.1 were used for the analysis. The results of the consistency analysis are shown in Table 6.

**Table 6.** Results of the consistency analysis.

| | Consistency Ratio | | | |
|---|---|---|---|---|
| **Recover** | **Use** | **Not in Use** | | |
| | <0.1 | >=0.1, <0.2 | >=0.2, <0.3 | >=0.3, <0.4 |
| 25 | 18 | 3 | 1 | 3 |

*4.2. Stakeholders' Overall SWOT-AHP Analysis*

By applying the quantitative AHP method to the SWOT output, stakeholder groups scored differently the importance of each factor. The factor priority weight reflects the relative importance of each factor within a SWOT category, while the global priority weights demonstrate the relative importance of each factor across all SWOT categories. It is important to note that the priority weights of the factors are relative values originating from pairwise comparisons made by the stakeholders (Table 7). In other words, factors with low priority values are less important, rather than not important, for successful integration of DEMP.

**Table 7.** SWOT-AHP priority weight for factors that can influence the implementation of the Digital Energy Management Platform.

| SWOT Factors and Categories | Factor Priority Weights | | | | | Global Priority Weights | | | | | | |
|---|---|---|---|---|---|---|---|---|---|---|---|---|
| | MCL | AG | ALGARDATA | AREAL | INFRAQUINTA | MCL | AG | ALGARDATA | AREAL | INFRAQUINTA | Overall | Rank |
| (S) Strengths | | | | | | 0.305 | 0.223 | 0.321 | 0.430 | 0.442 | 0.344 | 1 |
| S1: Early detection of anomalies, automatic suggestion of improvements and optimization of consumption | 0.096 | 0.087 | 0.092 | 0.097 | 0.460 | 0.029 | 0.019 | 0.029 | 0.042 | 0.204 | 0.065 | 7 |
| S2: Optimization of consumption and quantification of $CO_2$ emissions | 0.360 | 0.196 | 0.376 | 0.496 | 0.156 | 0.110 | 0.044 | 0.121 | 0.213 | 0.069 | 0.111 | 2 |
| S3: Production of reasoned and essential information for decision-making | 0.394 | 0.268 | 0.101 | 0.276 | 0.285 | 0.120 | 0.060 | 0.032 | 0.119 | 0.126 | 0.091 | 4 |
| S4: Potential for the use of energy resources | 0.150 | 0.449 | 0.432 | 0.131 | 0.099 | 0.046 | 0.100 | 0.139 | 0.056 | 0.044 | 0.077 | 5 |
| (W) Weaknesses | | | | | | 0.124 | 0.481 | 0.131 | 0.153 | 0.122 | 0.202 | 3 |
| W1: Difficulty in the interconnection of data from the energy supply company with the georeferencing of electricity meters | 0.125 | 0.133 | 0.398 | 0.114 | 0.383 | 0.015 | 0.064 | 0.052 | 0.017 | 0.047 | 0.039 | 12 |
| W2: Lack of specialized human resources | 0.442 | 0.453 | 0.129 | 0.440 | 0.103 | 0.055 | 0.218 | 0.017 | 0.067 | 0.012 | 0.074 | 6 |
| W3: Budget constraints | 0.053 | 0.279 | 0.093 | 0.334 | 0.429 | 0.007 | 0.134 | 0.012 | 0.051 | 0.052 | 0.051 | 9 |
| W4: Difficulty in interconnecting with other energy management systems | 0.381 | 0.135 | 0.380 | 0.112 | 0.085 | 0.047 | 0.065 | 0.050 | 0.017 | 0.010 | 0.038 | 13 |
| (O) Opportunities | | | | | | 0.454 | 0.087 | 0.468 | 0.308 | 0.372 | 0.338 | 2 |

**Table 7.** *Cont.*

| | | | | | | | | | | | | |
|---|---|---|---|---|---|---|---|---|---|---|---|---|
| O1: Existence of technologies capable of monitoring energy consumption in real time | 0.515 | 0.453 | 0.356 | 0.077 | 0.438 | 0.234 | 0.039 | 0.167 | 0.032 | 0.163 | 0.127 | 1 |
| O2: Increasing trend in electricity consumption | 0.237 | 0.091 | 0.087 | 0.078 | 0.104 | 0.108 | 0.008 | 0.041 | 0.032 | 0.039 | 0.0453 | 10 |
| O3: Existence of a legal framework that places particular emphasis on the implementation of energy efficiency measures and renewable energy sources | 0.123 | 0.355 | 0.104 | 0.322 | 0.088 | 0.056 | 0.031 | 0.049 | 0.132 | 0.033 | 0.060 | 8 |
| O4: Characteristics of the Loulé territory that favour the rational use of energy and the use of renewable energy sources | 0.126 | 0.102 | 0.454 | 0.273 | 0.370 | 0.057 | 0.009 | 0.212 | 0.112 | 0.138 | 0.106 | 3 |
| (T) Threats | | | | | | 0.118 | 0.209 | 0.080 | 0.109 | 0.064 | 0.116 | 4 |
| T1: Context of economic crisis | 0.380 | 0.103 | 0.103 | 0.091 | 0.089 | 0.045 | 0.022 | 0.008 | 0.010 | 0.006 | 0.018 | 16 |
| T2: Low Level of adherence to the awareness campaign about new energy management technologies among the population | 0.105 | 0.082 | 0.083 | 0.490 | 0.393 | 0.012 | 0.017 | 0.007 | 0.054 | 0.025 | 0.023 | 15 |
| T3: Difficulties in accessing investment and new energy management technologies | 0.454 | 0.390 | 0.425 | 0.088 | 0.444 | 0.054 | 0.082 | 0.034 | 0.010 | 0.028 | 0.041 | 11 |
| T4: High cost associated with the implementation of electrical equipment and use of renewable energy sources | 0.061 | 0.426 | 0.389 | 0.331 | 0.074 | 0.007 | 0.089 | 0.031 | 0.036 | 0.005 | 0.034 | 14 |

MCL: Group from Municipal Council of Loulé; AG: Academic Group; ALGARDATA: Group from the Computer Systems Company; INFRAQUINTA: Group from Quinta do Lago Infrastructure Company, EM; and AREAL: Algarve Regional Energy and Environment Agency.

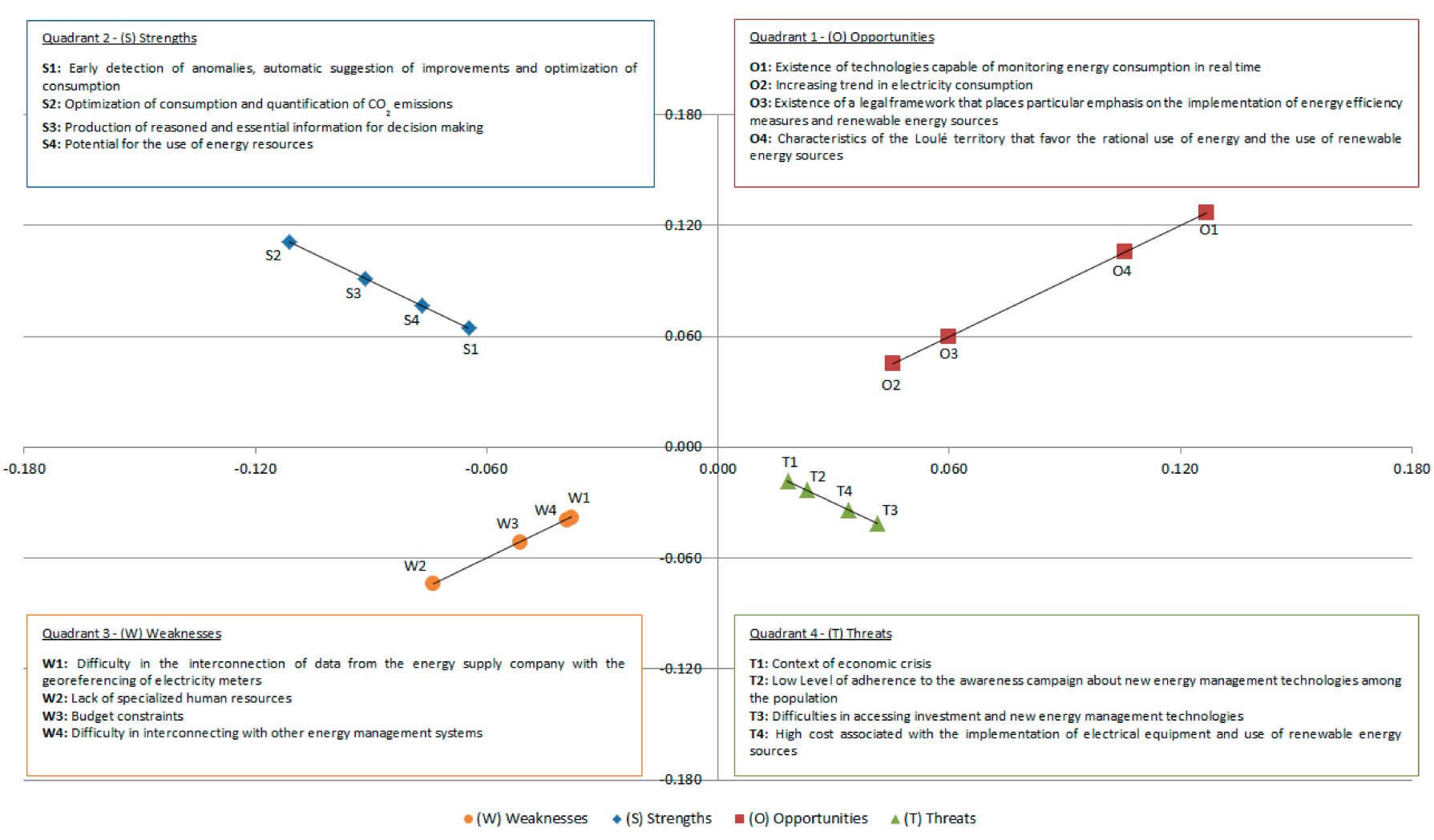

**Figure 4.** Perception map of responses from all stakeholders.

In addition, weights of Strength and Opportunity factors can be interpreted as positive of DEMP, while the weights of Weakness and Threat factors as negatives. For example, overall priority weights of 0.344 and 0.338 (column 11 of Table 7) suggest that Strengths and Opportunities factors of DEMP would account for 34.4% and 33.8%, respectively.

The SWOT-AHP analysis reveals that, on average, the stakeholders surveyed have a generally positive perception about the DEMP (Table 7). Therefore, the Strength category is the most important, with an average priority score of 0.344 (34.4%). The average priority score for all Opportunities combined was 0.338 (33.8%) while Weaknesses and Threats only obtained scores of 0.202 (20.2%) and 0.116 (11.6%), respectively.

A graph can be plotted based on the perception of all stakeholders combined, on the basis of priority weight given by all of them, as shown in Figure 4. Strength (S) and Opportunity (O), the positive categories, are plotted in quadrants 2 and 1, respectively, and Weakness (W) and Threat (T), the negative categories, are plotted in quadrants 3 and 4, respectively. The lengths of the straight lines on the graph represent the priority of the SWOT factors and their ratios to the total importance. The end points of the straight lines indicate the locations of the factors with the highest priority for each SWOT category. The remaining factors are plotted on the lines according to the values of their priority's weights. It is important to note that when scores are disaggregated and analysed for different groups, perceptions of each group differ, and so do the scores.

The Strength factors are in the order S2 > S3 > S4 > S1. Based on the scores, stakeholders consider the optimization of consumption and quantification of $CO_2$ emissions (S2) the most important Strength for the municipality of Loulé. According to the experts, the integration of the DEMP will be an essential tool to achieve a 40% reduction in energy consumption and 40% of the respective $CO_2$ emissions by 2030 in the municipality of Loulé. The added solutions will reduce dependence on fossil fuels and consequently $CO_2$ emissions.

S2 is followed by S3, which is associated with production of reasoned and essential information for decision-making. Experts believe that if it is possible to monitor consumption and $CO_2$ emissions, it will be possible to implement mitigation actions, namely initiatives to improve energy efficiency and increase renewable production with the potential to reduce greenhouse gas emissions. Aspects that influence sustainability policies were mentioned as specific energy and environmental challenges, types of decision-making processes, and accessibility of resources [51]. According to Lepenioti et al. [52] with the increase in substantial amounts of data within organizations, it is necessary to implement mechanisms that allow organizations to improve decision-making and process effectiveness, providing actions for better management. However, energy management strategies are also needed to support the efficient integration of renewable energy sources into the existing energy networks [53].

The Weakness factors are in the order W2 > W3 > W4 > W1. W2 is associated with the lack of specialized human resources. Stakeholders consider W2 the most limiting internal barrier, noting that the availability of specialized and ready-to-work human resources in the public sector is key to the success of the DEMP integration. Each municipal technician needs clear guidelines on how the data should be read, which data should be considered, and who is responsible. Zekić-Sušac et al. [51] stated the importance of an energy manager who is able to interact and contact with providers of energy services, as well as to define the most appropriate technological and organizational solutions to optimize energy resources.

The Opportunity factors are in the order O1 > O4 > O3 > O2. O1 is the most important factor. Experts assert that with the existence of technologies capable of monitoring energy consumption in real time, allowing data to be processed and analysed by DSRN technicians, more rigorous and precise studies can be performed. This result is similar to what was mentioned by Zekić-Sušac et al. [51]. These authors indicated that intelligent energy systems will be able to predict energy consumption and costs with the aim to assist decisionmakers in the public sector. Eicker et al. [11] referred that the use of a platform is critical for smart, sustainable, and resilient city planning, operation, and maintenance.

The Threats are in the order T3 > T4 > T2 > T1. T3 is rated the biggest Threat to platform integration. The use of DEMP in the public sector is a prominent issue in the context of smart cities due to the increase in the energy consumption of buildings, especially public buildings such as education, health, government, and other public institutions with high frequency of use. However, experts consider that there are difficulties in accessing investments to new energy management technologies in the public sector. Halkos and Gkampoura [54] indicated that the economic crisis of a region is directly reflected in energy efficiency, in the ease of access to new technologies and renewable energy sources.

All four Strength (S) factors, three Opportunity (O) factors, two Weakness (W) factors and one Threat (T) factor are ranked as the top factors. The priority of the factors within the SWOT categories is indicated by the priority weight values. The factor with the greatest priority values within each SWOT category are included among the factors with the eight highest priority values. The Figure 5 presents the top eight factors in order of importance.

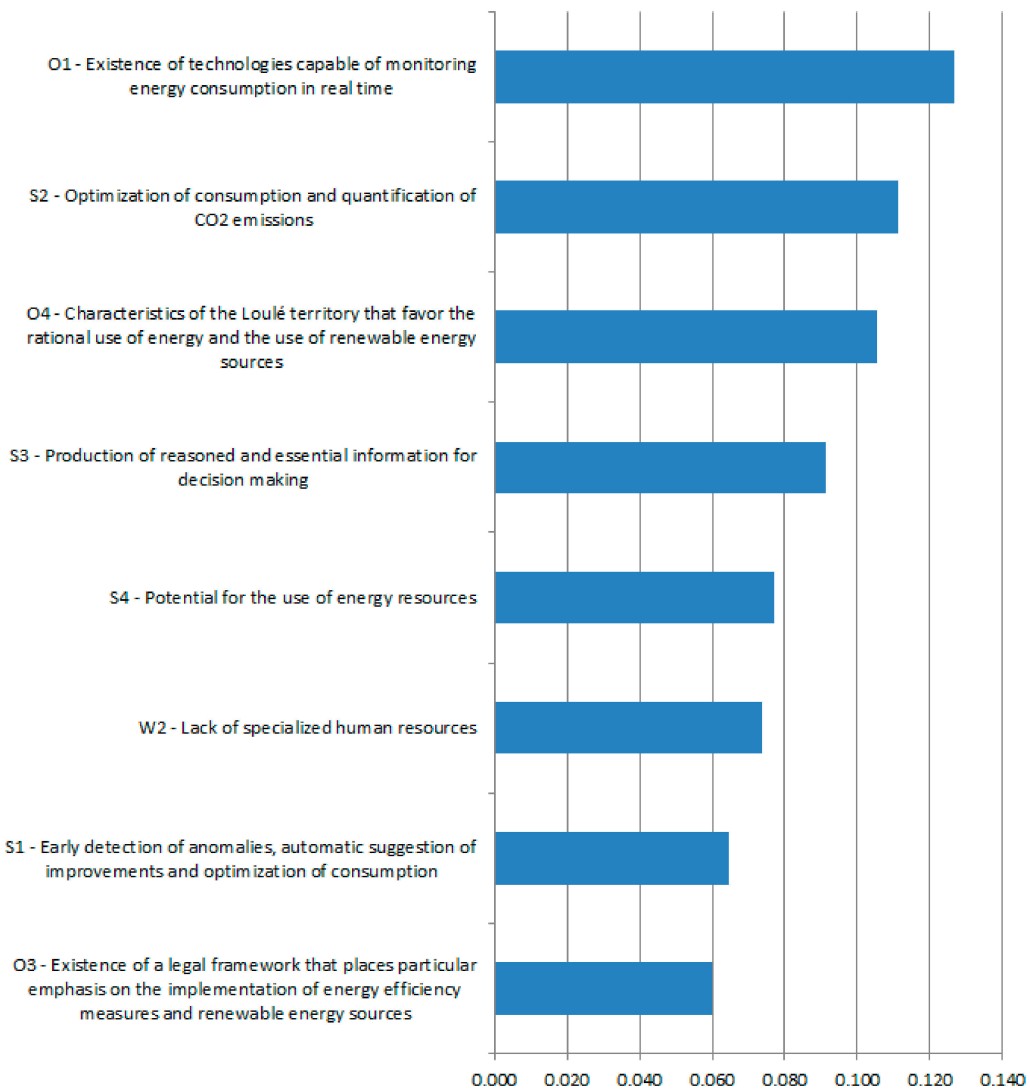

**Figure 5.** Top eight factors of global priority within the SWOT categories.

This comprehensive analysis shows that Opportunities O1, O4, and O3 and Strengths S2, S3, S4, and S1 should be leveraged and that Weakness W2 should be dealt with. The findings also indicate that municipality of Loulé should utilize Opportunity O1 and Strength S2 to integrate the new DEMP.

### 4.3. Diversity in the Perception of Stakeholder Groups

The different stakeholder groups have slight variations in perceptions of SWOT factors and categories. Figure 6 shows the differences among stakeholder groups in terms of the global priority of SWOT categories, e.g., Strengths versus Weaknesses. The overall results show that the Strengths category was considered to contain the most important factors for the implementation of the DEMP in the municipality of Loulé.

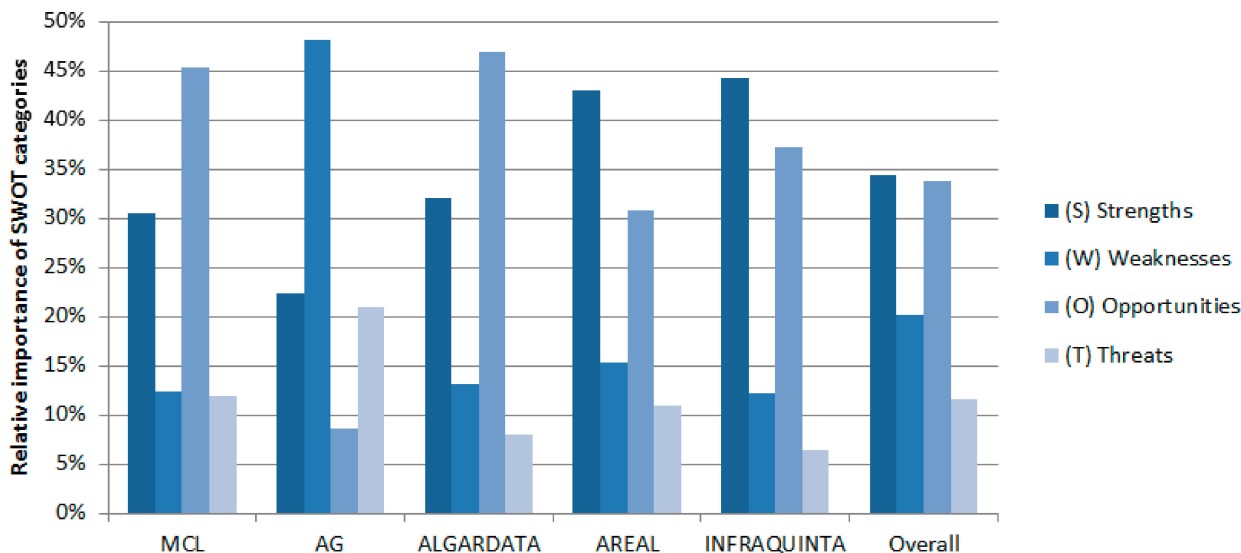

**Figure 6.** Stakeholder group perceptions of the importance of SWOT categories for to integrate of the Digital Energy Management Platform in the municipality of Loulé. MCL: Group from Municipal Council of Loulé; AG: Academic Group; ALGARDATA: Group from the Computer Systems Company; INFRAQUINTA: Group from Quinta do Lago Infrastructure Company, EM; AREAL: Algarve Regional Energy and Environment Agency.

Stakeholders from the Municipal Council of Loulé (MCL) and the Computer Systems Company group (ALGARDATA) consider the Opportunity category as the most relevant. since these two entities work together in the development and updating of the DEMP. ALGARDATA develops a complete and intelligent database and MCL implements it, enabling impact assessment through planning and monitoring performance indicators to reduce large energy consumers and is strongly committed to climate change mitigation. The groups from the Algarve Regional Agency for Energy and Environment (AREAL) and Quinta do Lago Infrastructure Company (INFRAQUINTA) present the Strength category as the most important. These companies have strategic goals, mission, vision, and values in common to achieve a digital energy transformation, and consider that the platform is easy and advantageous to adjust and use. However, the Academic Group (AG) considers the categories Weakness and Threats relatively more important than other stakeholder groups, because of the difficulty in the access to new energy investments, associated with changes in public sector management. These may be decisive factors for the effective implementation, monitoring, and future updates of DEMP.

In terms of the individual factors (Figure 7), prioritisation varies between stakeholder groups. Based on the weights assigned to each factor, the various perspectives of the stakeholder groups were consolidated into three key objectives for the integration of the platform in the municipality of Loulé: (i) smart database, (ii) better management service in the public sector, and (iii) ease of optimisation. The smart database objective provides a platform that provides appropriate information from the Loulé territory, allowing online and real-time access to energy data and possible interconnection with data from new renewable energy sources. This objective was highlighted by the MCL and ALGARDATA expert groups, with a relative importance of 23.4% for factor O1 existence of technologies capable

of monitoring energy consumption in real time, and 21.2% for factor O4 characteristics of the Loulé territory that favour the rational use of energy and the use of renewable energy sources. The second objective, better management service in the public sector, is relative to the implementation of a tool that allows financial management and qualified human resources. This objective was identified by the AG expert groups, with a relative importance of 21.8% to factor W2 lack of specialized human resources, and 8.9% to factor T4 high cost associated with the implementation of electrical equipment and use of renewable energy sources. The last key objective, ease of optimization, highlights the need of internal processes optimisation, through an intelligent platform that allows information to be managed appropriately, facilitating decision-making in sectors with high energy consumption and promoting a decrease in $CO_2$ emissions. This objective was highlighted by the AREAL and INFRAQUINTA expert groups, with a relative importance of 21.3% for factor S2 optimization of consumption and quantification of $CO_2$ emissions, and 20.4% for factor S1 early detection of anomalies, automatic suggestions of improvements and optimization of consumption, respectively.

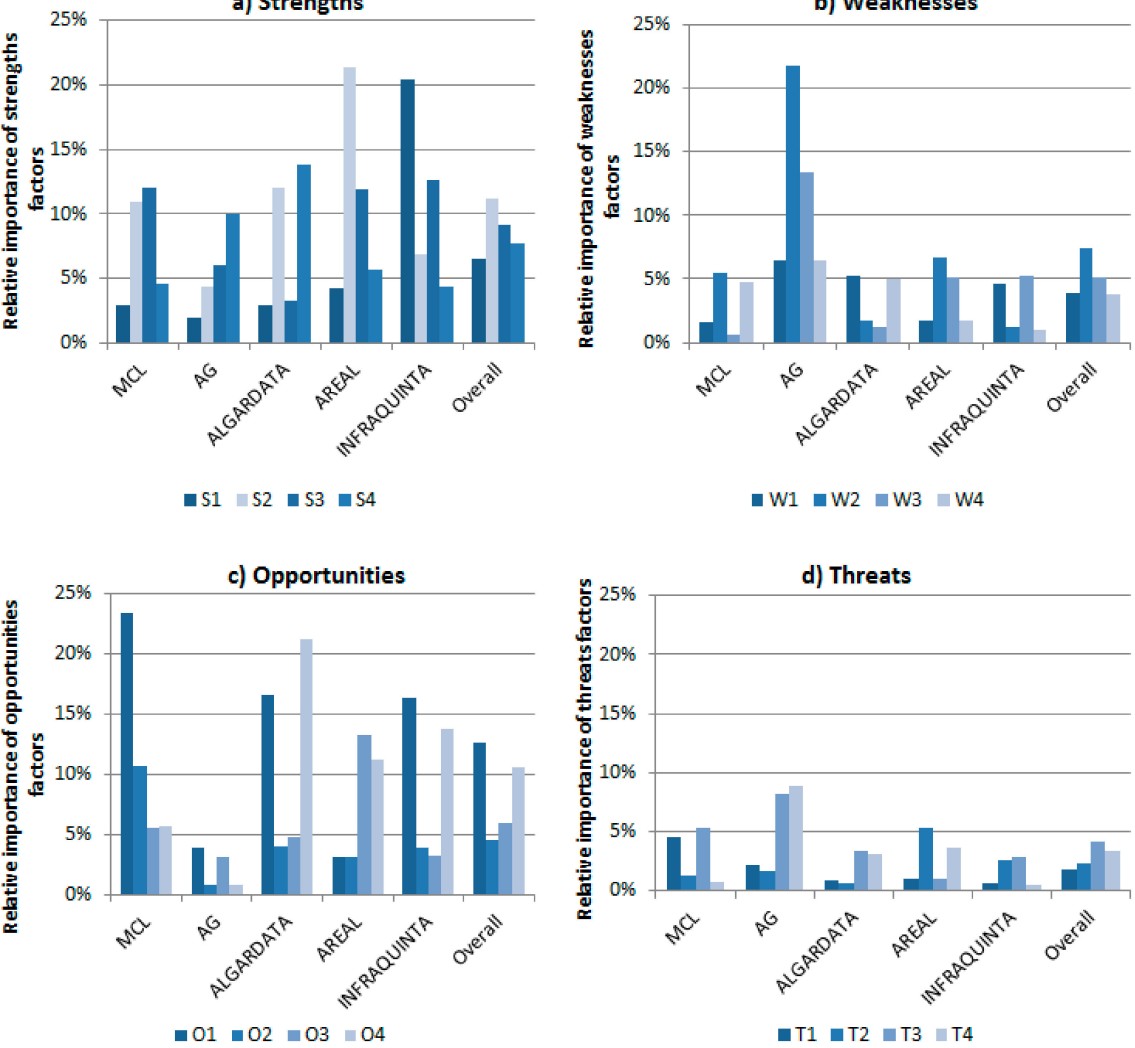

**Figure 7.** Perceptions of stakeholders of the importance of the factors in each category: (**a**) Strengths; (**b**) Weaknesses; (**c**) Opportunities, and (**d**) Threats. MCL: Group from Municipal Council of Loulé; AG: Academic Group; ALGARDATA: Group from the Computer Systems Company; INFRAQUINTA: Group from Quinta do Lago Infrastructure Company, EM; AREAL: Algarve Regional Energy and Environment Agency. Note: for a list of descriptions of each factor, see Tables 5 and 7.

In summary, the experts state that the municipality of Loulé has contributed to a smart digital transformation, namely by developing and implementing technological applications, with the platform being an innovative tool for a better energy management, reinforcing the climate change mitigation, and contributing to the development of smart cities.

### 4.4. Strategy Formulation Using a TOWS Matrix

The TOWS Matrix essentially comprises, Strengths-Opportunities (SO), Strengths-Threats (ST), Weaknesses-Opportunities (WO), and Weaknesses-Threats (WT) strategies [55].

After the ranking of the identified SWOT factors, a TOWS Matrix was developed for strategy formulation. The TOWS Matrix approach consists of a framework for developing possible national strategies by analysing the Strengths and Weaknesses of energy management and integrating them with the Opportunities and Threats for DEMP integration. The main purpose of the identified strategies is to help the integration of DEMP, especially to reduce energy consumption, costs, and $CO_2$ emissions by the municipality of Loulé. To develop the strategy, a thorough analysis of Weaknesses (W) and Strengths (S) is essential, which operates in a larger external environment that sets up Threats (T) but also offers Opportunities (O) to the system. The four factors mentioned can become the base for the development of four different strategies. An SO (or max-max) strategy which uses Strengths to take advantage of Opportunities is considered the most favourable, i.e., it takes advantage of Opportunities using its Strengths. Alternatively, an ST (or max-min) strategy aims to maximise the Strengths of DEMP integration by minimising the Threats. Similarly, a WO (or min-max) strategy overcomes Weaknesses by using external Opportunities. A WT (or min-min) strategy is the least favourable in the TOWS Matrix, this occurs when external Threats and internal Weaknesses may hinder the continuation of the municipality of Loulé in DEMP integration. This strategy aims to minimise both internal Weaknesses and external Threats [49]. Table 8 describes the four strategies. Tags in parentheses indicate which SWOT factor will be impacted by the strategy.

According to the results presented in Table 8, a strategic quadrilateral was built, which can directly reflect the orientation of the best strategy for DEMP implementation (Figure 8).

Based on the comprehensive SWOT-AHP analysis, possible strategies for the implementation of the DEMP are Strength-Opportunity (SO) and Weakness-Opportunity (WO) strategies, with a prevalence of 34% and 27%, respectively (Figure 8). These strategies are derived based on the global priorities of the SWOT categories, as shown in Table 8. Strength-Threat (ST) and Weakness-Threat (WT) strategies were not suggested because it was considered only the top eight SWOT factors based on their priority's weights.

#### 4.4.1. Strength-Opportunity (SO) Strategy

O1 (existence of technologies capable of monitoring energy consumption in real time), S2 (optimization of consumption and quantification of $CO_2$ emissions), O4 (characteristics of the Loulé territory that favour the rational use of energy and the use of renewable energy sources), S3 (production of reasoned and essential information for decision-making), S4 (potential for the use of energy resources), S1 (early detection of anomalies, automatic suggestion of improvements and optimization of consumption) and O3 (existence of a legal framework that places particular emphasis on the implementation of energy efficiency measures and renewable energy sources) are included in this strategy.

**Table 8.** Strategic formulation using the TOWS Matrix.

| Internal Elements | S | W |
|---|---|---|
| **External Elements** | S1: Early detection of anomalies, automatic suggestion of improvements and optimization of consumption (0.065)<br>S2: Optimization of consumption and quantification of $CO_2$ emissions (0.111)<br>S3: Production of reasoned and essential information for decision-making (0.091)<br>S4: Potential for the use of energy resources (0.077) | W1: Difficulty in the interconnection of data from the energy supply company with the georeferencing of electricity meters (0.039)<br>W2: Lack of specialized human resources (0.074)<br>W3: Budget constraints (0.051)<br>W4: Difficulty in interconnecting with other energy management systems (0.038) |
| O | O1: Existence of technologies capable of monitoring energy consumption in real time (0.125)<br>O2: Increasing trend in electricity consumption (0.044)<br>O3: Existence of a legal framework that places particular emphasis on the implementation of energy efficiency measures and renewable energy sources (0.053)<br>O4: Characteristics of the Loulé territory that favour the rational use of energy and the use of renewable energy sources (0.100) | **SO Strategy**<br><br>Promoting the use of a platform that allows real-time management of electricity consumption, $CO_2$ emissions, and costs in the long term.<br>S(S1,S2,S3,S4) + O(O1,O4,O3) | **WO Strategy**<br><br>Increasing the skilled workforce and promote specialized training in technologies capable of monitoring energy consumption.<br>W(W2) + O(O1,O4,O3) |
| T | T1: Context of economic crisis (0.018)<br>T2: Low level of adherence to the awareness campaign about new energy management technologies among the population (0.023)<br>T3: Difficulties in accessing investment and new energy management technologies (0.041)<br>T4: High cost associated with the implementation of electrical equipment and use of renewable energy sources (0.034) | **ST Strategy**<br><br>NA | **WT Strategy**<br><br>NA |

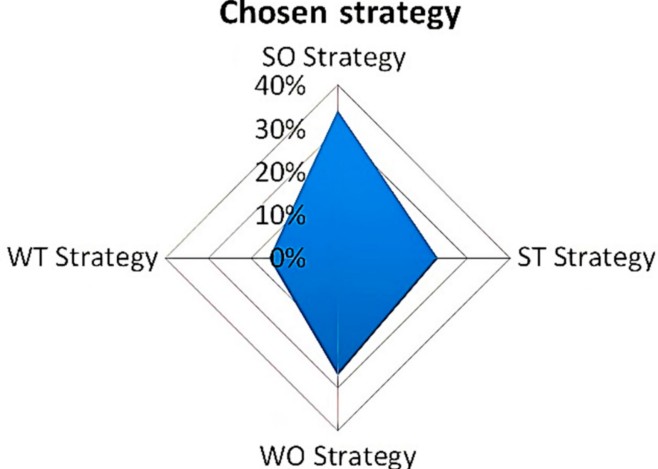

**Figure 8.** Strategic quadrilateral of DEMP integrate.

First, S2, the key factor within the Strength category, suggests that the DEMP should optimize of consumption and quantification of $CO_2$ emissions. Experts state that this strategy allows the municipality to play a more active role around a decentralized national energy system, with the establishment of smart energy networks, the growing integration of electric mobility and the emergence of regulation of renewable energy communities. Data on energy consumption, energy costs and $CO_2$ emissions were obtained directly from the billing systems of the municipality's energy suppliers. In each invoice, the energy supplier also includes information about the energy source for the period billed (solar, wind, hydroelectric, etc.) and the respective $CO_2$ emissions [12]. According to experts, this strategy can be a new mechanism for analysing and optimising urban energy infrastructure. Building, commerce, and industry as well as the transport sector are in the focus of the efficiency and renewable supply analysis. Zekić-Sušac et al. [51] described that it is necessary to implement energy management strategies in the public sector, with the objective of helping to control $CO_2$ emissions.

Further, a Strength-Opportunity strategy that combines S1 and O1 can be proposed for early detection of anomalies, optimisation of the real time consumption, and to automatically receive suggestions of system improvements. Thus, DEMP allows not only a global perception of the general situation of consumption, $CO_2$ emissions and electricity costs, but also contributes to identify priority actions that should be taken, helping in decision-making in municipal management. Eicker et al. [11] referred that the platforms can provide real time data on urban infrastructure performance, air quality, thermal comfort, and more. The International Energy Agency [56] stated that this strategy can provide greater clarity for decisionmakers and reported that new digital technologies are set to make energy systems more connected, intelligent, efficient, reliable, and sustainable.

### 4.4.2. Weakness-Opportunity (WO) Strategies

W2 (Lack of specialized human resources) has high priority among the Weakness factors (W). There are difficulties in overcoming the emerging needs and in promoting a better allocation of resources because the lack of human resources which promote low productivity, delay tasks, and consequently economic loss. Experts indicate that human resources departments should systematically apply practices that ensure motivation and satisfaction of workers, promoting the continuous improvement of individual and collective performance through the development of awareness and training actions in the area of energy management. With the lack of specialized human resources and increasing demand for reliable energy data in the day-to-day operation, the integration of a DEMP into the public sector is a major challenge.

In a recent study carried out by Hassan et al. [57] on the diagnosis of issues and challenges in data integration implementation in the public sector, human resource skills

and competency were mentioned. Lack of skills in both technical and non-technical areas will become a big challenge. Not only skills and knowledge are important, but also the development of a culture of information shared between public sector staff is a challenge to be faced [57–59].

The increased motivation of workers provides innovation and creativity, which translates into the improvement of internal processes and the achievement of service objectives. This responds to the expectations and satisfaction of the users through service quality and favourable financial results. According to Organization for Economic Cooperation and Development [60], the lack of qualified personnel is one of the factors that hinders innovation activity in the field of knowledge.

## 5. Conclusions

This study offers the first application of the SWOT-AHP technique in appraising factors that can influence the implementation of a Digital Energy Management Platform (DEMP) in the public sector. In this context, since the municipality of Loulé has a deeply rooted institutional structure with many technological advances in terms of energy management, this technique can be extended to predict energy efficiency level, consumption of propane gas, fuel, and other energy resources. In this study, key stakeholders who influence the collection, monitoring, analysis of data, and system innovations for energy were consulted.

The Strengths and Weaknesses of the existing system and the Opportunities and Threats found in relation to the external environment were evaluated and it is proposed that this new DEMP should be adopted by decisionmakers to optimize the processing and treatment of energy data in the municipality of Loulé. The results show the following ranking of each SWOT category priority: Strengths (34.4%), Opportunities (33.8%), Weaknesses (20.2%), and Threats (11.6%). The most prominent Opportunity for the implementation of DEMP is the ability of this technology to provide real time energy monitoring and storage and processing. This Opportunity will address the growing need for electricity in the near future, cost rationalization, and a desire for cleaner sources of electricity.

This study shows that the Opportunity factor with the highest overall score by stakeholders was the existence of technologies capable of monitoring energy consumption in real time, being approximately 12.7%, right after optimization of consumption and quantification of $CO_2$ emissions which was the Strength factor with the highest overall score of approximately 11.1%. The DEMP will assess which installation is spending the most and, confirm/evaluate the estimates from smart meters, check the contracted power, the need to apply capacitors, as well as provide timely data to each municipal department in order to make the best management decision. On the other hand, the biggest Weakness of the DEMP implementation in the municipality of Loulé is the lack of specialized human resources, with an overall score from stakeholders of 7.4%, meeting the need to hire and promote specialized training on the use of DEMP. Regarding the existing Threats, the one that prevailed in the score provided by the stakeholders was difficulties in access to investment and new energy management technologies, obtaining an overall score of 4.1%. Recently there has been the appearance of new investments, but these are not yet sufficient.

Based on stakeholder's perceptions, this study shows that the appropriate strategies for integrating DEMP in the municipality of Loulé are SO strategies, which use Strengths and Opportunities, and WO strategies, which take advantage of Opportunities while dealing with Weaknesses, achieving a prevalence compared with the other strategies of 34% and 27%, respectively. After the definition of strategies for DEMP integration, the level for data interpretation and visualization will support users in making decisions on future actions that will enable benefits at a micro level (in each building), as well as at a macro level (for whole public sector). This digital platform allows the improvement of energy supply in the future and mitigate its environmental impact, namely through the development and implementation of more energy efficient and low carbon technologies.

Additional investigation is required to understand how the strategies can be applied to the Digital Energy Management Platform based on the derived factors within the SWOT categories.

**Author Contributions:** Conceptualization, D.G. and P.M. (Paula Mendes); methodology, D.G. and P.M. (Paula Mendes); data collection, preparation and curation, D.G., P.M. (Pedro Murta) and N.C.; writing—original draft preparation, D.G.; writing—review and editing, M.R.T.; supervision, M.R.T.; project administration, P.M. (Paula Mendes). All authors have read and agreed to the published version of the manuscript.

**Funding:** This research received no external funding.

**Institutional Review Board Statement:** Not applicable.

**Informed Consent Statement:** Not applicable.

**Data Availability Statement:** Not applicable.

**Acknowledgments:** The authors acknowledge the municipality of Loulé for the permission to use the data. Special acknowledgements to the Mayor of Loulé, Vitor Aleixo, and technicians Ana Paula Neves and Ilda Lima for supporting the data analysis, and Francisco Sousa, involved in the Digital Energy Management Platform support. We also gratefully acknowledge the involvement of all stakeholders (UAlg; ALGARDATA; AREAL and INFRAQUINTA) in completing the questionnaire for this study.

**Conflicts of Interest:** The authors declare no conflict of interest.

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
