# Peer review of "Stakeholders’ Perceptions of New Digital Energy Management Platform in Municipality of Loulé, Southern Portugal: A SWOT-AHP Analysis"

_sustainability, doi:10.3390/su14031445_

Round 1

Reviewer 1 Report

This is an interesting paper that seeks to develop a multi-stakeholder analysis to identify the best strategies for 13 the integration of a new Digital Energy Management Platform. It is worthy to be published however before it will be recommended for publication, the following comments must be addressed by the authors.

  1. There are minor grammatical errors that run through the paper, authors are therefore requested to take a second look at the English and correct such errors.
  2. There is the need to strengthen the literature review with recent papers that combined the SWOT and AHP in similar manner. Such recent papers can be found in sciencedirect and MDPI journals.
  3. More importantly authors failed to explain clearly how they consolidated the various views of the experts consulted. This is very key in such analysis since the AHP is key in assigning the various weights to the various criteria identified. 
  4. Which of the AHP's did they use in the consolidation? Is it GM or by consensus or arithmetric? Each of these methods have their unique mathematical relations that need to be used. Authors must come clear and tell us how the consolidation was done and provide the appropriate mathematical relations. This is very important. As it stands it looks confusing and problematic.
  5. The abstract should be improved with more quantitatives.

Reviewer 2 Report

Introduction:

The study provides a well structured hybrid SWOT-AHP-TOWS Model. It tackles an important issue to provide a Digital Energy Management Platform and considers on-the-ground case study to examine the proposed model. 

Comments:

(1) Line 128:

Is the growing consumption the only driver for the intelligent energy management platform?
Other relevant reasons might be also considered, such as: 
1- The integration of RESs (if any), which requires higher level of load prediction
2- The improvement of the grid to a smart grid.
3- Support energy efficiency efforts through identifying the high consuming sectors.

(2) Line 159:

Good representing panelist and good presentation of their background. It would be coherent to elaborate in the expected input/deriving reason to select the representatives of each sector (e.g. MCL due to their background on the demand and ability of providing prediction of its growth). 

(3) Line 182:

In Figure (1), the  satisfactory results depend on what? consistency ratio? Clarify the deciding factor(s).

(4) Line 217:

The AHP is one of the most widely applied MCDA due to (one reason is mentioned here and another in a next sentence). It would be better to list the reasons such as:

1- can identify the contradictory preferences through consistency ratio to be avoided and re-evaluated, 
2- Pairwise comparison of elements (mentioned in next sentence) 
3- Ease of application with experts
4- Synthesis of the provided judgments over different levels of the hierarchy.

(5) Line 259:

It is important to provide a Figure of the AHP hierarchy with the different levels of criteria/sub-criteria to help the reader to understand the following process.

(6) Line 350:

It is a common practice in AHP studies to explain how are the decision factors defined. In this case, as the applications of SWOT-AHP-TOWS stated to be limited in the energy management domain, it is important to explain how are the decision factors determined.

(7) Lines 362 and 397:

It is good to present quantitative analysis of CR as the authors did in Table (5). Also, the discussion of the results after the Perception map of responses from all stakeholders Figure is comprehensive. Well done!

(8) Line 445:

Why 8 factors specifically? (Weight of 0.6 and more?)

(9) Line 460:

The Academic Group is the only group that gives higher weight for Threats? This looks interesting and possible reasons could be indicated here.

Spelling check, Figures resolution, and Captions

(1) Line 147:

for organisations (to) know how.

(2) Lines 239, 279, and 520:

The resolution of Figure (2), Equation (1), and Figure (8) need to be improved. 

(1) Line 394:

Figure Number should be (4)

Round 2

Reviewer 1 Report

Can be accepted now